# Near-Infrared Photoimmunotherapy for Thoracic Cancers: A Translational Perspective

**DOI:** 10.3390/biomedicines10071662

**Published:** 2022-07-11

**Authors:** Kohei Matsuoka, Mizuki Yamada, Mitsuo Sato, Kazuhide Sato

**Affiliations:** 1Department of Integrated Health Sciences, Nagoya University Graduate School of Medicine, 1-1-20, Daiko, Higashi-ku, Nagoya 461-8673, Japan; matsuoka.kouhei.h4@f.mail.nagoya-u.ac.jp (K.M.); yamada.mizuki.s3@s.mail.nagoya-u.ac.jp (M.Y.); msato@met.nagoya-u.ac.jp (M.S.); 2Respiratory Medicine, Nagoya University Graduate School of Medicine, Tsurumai-cho 65, Chikusa-ku, Nagoya 466-8550, Japan; 3B3 Unit, Advanced Analytical and Diagnostic Imaging Center (AADIC)/Medical Engineering Unit (MEU), Nagoya University Institute for Advanced Research, Nagoya University, Tsurumai-cho 65, Showa-ku, Nagoya 466-8550, Japan; 4FOREST-Souhatsu, CREST, JST, Goban-cho 7, Chiyoda-ku, Tokyo 102-0076, Japan

**Keywords:** near-infrared photoimmunotherapy, NIR-PIT, phototherapy, thoracic tumor, lung cancer, target antigens

## Abstract

The conventional treatment of thoracic tumors includes surgery, anticancer drugs, radiation, and cancer immunotherapy. Light therapy for thoracic tumors has long been used as an alternative; conventional light therapy also called photodynamic therapy (PDT) has been used mainly for early-stage lung cancer. Recently, near-infrared photoimmunotherapy (NIR-PIT), which is a completely different concept from conventional PDT, has been developed and approved in Japan for the treatment of recurrent and previously treated head and neck cancer because of its specificity and effectiveness. NIR-PIT can apply to any target by changing to different antigens. In recent years, it has become clear that various specific and promising targets are highly expressed in thoracic tumors. In combination with these various specific targets, NIR-PIT is expected to be an ideal therapeutic approach for thoracic tumors. Additionally, techniques are being developed to further develop NIR-PIT for clinical practice. In this review, NIR-PIT is introduced, and its potential therapeutic applications for thoracic cancers are described.

## 1. Introduction

Thoracic tumors, including lung cancer and malignant pleural mesothelioma (MPM), are the most lethal cancers worldwide [1,2]. To overcome thoracic cancer, NIR-PIT may be an ideal treatment and can lead to significant improvements in treatment outcomes. There are four conventional treatments for cancer: surgery, radiation therapy, chemotherapy, and immunotherapy. However, these existing treatments injure not only cancer cells but also the surrounding normal cells, tissues, and organs. In particular, these treatments may impact the vital organs such as the liver, the heart and the thoracic aorta, and bone marrow. The highly specific and effective NIR-PIT can overcome this problem because of its combined and multidisciplinary approach. In addition, NIR-light used in NIR-PIT is transmitted easily through the tissue filled by air. NIR light with a wavelength range of 650–900 nm is not readily absorbed by water or hemoglobin, and penetration through the tissue is maximal [3,4,5,6]. Therefore, the NIR wavelength of 690 nm used in NIR-PIT is thought to be a suitable wavelength for human therapy and is not toxic to the body.

Historically, thoracic tumors have been treated with light. The first bronchoscopic PDT was carried out in a patient with early central lung cancer by Hyata et al. with encouraging results. Since then about 1000 cases of ECLC have received PDT [7]. Additionally, O.J. Balchum and colleagues used PDT to treat patients with lung cancer. All of these studies showed promising responses in early-stage patients, so PDT was recommended for patients with early-stage cancers that were inoperable, due to other complications [8]. However, PDT, which is mediated ROS, can damage not only target cancer cells but also normal cells. On the other hand, NIR-PIT is highly specific and effective as a result of its combined and multidisciplinary approach.

In this review, we describe the potential application of NIR-PIT for thoracic cancer and discuss this perspective in detail.

## 2. Summary

NIR-PIT is an ultra-specific and effective cancer photoimmunotherapy method that can be used to treat cancers throughout the body. NIR-PIT utilizes an antibody-photon absorbent conjugate of IRDye700DX (IR700), a near-infrared water-soluble silicon phthalocyanine derivative, and an mAb that targets surface antigens expressed on cancer cells [9,10]. Irradiation of the targeted tumor site with near-infrared light (approximately 690 nm) activates the conjugates (Figure 1①,②), induces aggregation of antibody and antigen proteins, and causes cell death (Figure 1③). This two-step targeted selection (antibody and light) provides a double level of specificity, which is not found in conventional cancer treatments. 

The mechanism of NIR-PIT was elucidated after the launch of the first phase III study [10]. Specifically, when the conjugates were irradiated with near-infrared (NIR) light in the presence of sufficient electron donors, the hydrophilic side chains of the IR700 molecule (silanol) dissociated through a photochemical ligand reaction, and the remaining structure, including the antibody, rapidly became hydrophobic and aggregated. At that time, the antibody binding the surface antigen on the cell membrane of the tumor also aggregated, forcing cells to break. Fluorescence imaging could track where the conjugates were distributed and how well mAb-IR700 conjugates reacted, and the increase in IR700-fluorescence indicated that the mAb-IR700 was well connected to the target, while the decrease could confirm the photochemical reactions caused by NIR light irradiation and correlated with the treatment effect. The development of NIR-PIT was interdisciplinary, involving biology, physics, and chemistry, making use of the advantages of each discipline.

The unique feature that makes IR700 useful and valuable is its hydrophilicity. It never permeates the lipid bilayers of cell membranes and does not change the pharmacokinetics of mAbs in vivo, unlike hydrophobic photosensitizers in conventional photo-based cancer therapies such as PDT. In addition, mAb-IR700 conjugates mainly bind to the membrane of targeted tumor cells after intravenous injection and destroy the cell membrane with NIR light. Moreover, it was elucidated that reactive oxidative species (ROS) were not included in the main mechanism for NIR-PIT, since the cell death reaction in NIR-PIT proceeded even after the cell function was stopped at 4 °C, and the inhibition of cell death was not sufficiently effective even when oxidative stress inhibitors (free radical scavengers) were added. Moreover, the mass spectrometric analysis revealed that photochemical reaction is the trigger of the cell death induced by NIR-PIT [11]. The pharmacodynamics of the photosensitizer and pharmacokinetics of the conjugates in vivo are completely different between traditional PDT and NIR-PIT, which makes NIR-PIT a new photo-based modality [10].

NIR-PIT not only causes physical destruction of cells during NIR irradiation, but also induces secondary immunogenic cell death (ICD) due to damage-associated molecular patterns (DAMPs), and the release of entities such as adenosine triphosphate (ATP), calreticulin (CRT), and high mobility group box 1 (HMGB1) [12]. ICD has also been reported to occur with other radiotherapies, chemotherapy, and PDT, but these treatments themselves have a certain degree of toxicity to immune cells. NIR-PIT especially benefits from ICD [13,14,15], because this immunogenic cell death is not limited to the irradiated tumor but also has an effect on tumors elsewhere and on metastatic tumors. This combination of NIR-PIT and immune checkpoint inhibitors is currently being tested in phase II trials worldwide, but these do not include thoracic cancer (https://rakuten-med.com/us/pipeline/ accessed on 9 July 2022).

## 3. SUPR Effect

Cancer treatment with drugs has been considered an inappropriate therapy in vivo largely because of difficulties in drug retention in affected areas, such as vascular heterogeneity and high interstitial pressure [16,17]. Tumor tissue has increased permeability and retention of nanoparticles due to vascular abundance and lack of lymphoid tissue, referred to as the EPR effect. Conventional anti-cancer nanopharmaceuticals aim to take advantage of this effect to accumulate in tumors [18,19,20]. Although the EPR effect improves drug delivery to tumors compared to normal tissue, the effect is not significant and only low concentrations of nanodrugs reach and accumulate in tumors. The best-known nanopharmaceuticals are liposomal formulations such as Doxil and DaunoXome, both of which are as effective as small molecule formulations, but require more frequent administration and are impractical [21,22].

NIR PIT helps deliver nano-sized drugs desired to stay in the tumor; NIR-PIT is a very specific therapy that ruptures only the cancer cells labeled with the conjugate. Administered intravenously into the body as a drug, mAb-IR700 reacts from the vascular periphery to bind to target cells and specifically destroy them. The rapid rupture of surrounding cancer cells causes 10~20 times the EPR effect of vasodilation, increased blood flow, decreased blood flow velocity in the tumor, and increased vascular permeability, called super permeabilization and retention (SUPR) [23,24]. In other words, NIR-PIT can contribute to direct therapy and enhanced delivery of nanodrugs.

## 4. The Versatile Targets: EGFR, HER2, CD44, and CEA

NIR-PIT can be used to change the treatment pathway by creating mAb-IR700 using antibodies that target different ligands. As an entry point of research, it is simple to apply targets that have already been used in clinical practice (Table 1).

### 4.1. EGFR

Epidermal growth factor receptor (EGFR) is a transmembrane tyrosine kinase receptor belonging to the erythroblastosis oncogene B (ErbB) family [25]. EGFR overexpression is associated with poor prognosis in several tumor types, including thoracic cancer [26]. Physiologically, EGFR regulates epithelial tissue development and homeostasis. EGFR mutation and/or overexpression has been observed in several human cancers, and EGFR-targeted therapy has become a routine part of the treatment of several cancers. EGFR is considered a suitable target for early clinical trials of NIR-PIT. Several studies have confirmed the therapeutic efficacy of NIR-PIT targeting EGFR in in vivo models of several types of cancer, including lung cancer [27,28,29,30,31,32,33,34,35]. Cetuximab-IR700, a chimeric IgG1 monoclonal antibody against EGFR, was approved under certain conditions, such as limiting its use to HNSCC and was the first EGFR-targeted NIR-PIT drug to be registered for clinical use in Japan in 2020 [36]. For 30 patients enrolled in Phase 2 of the clinical practice (RM-1929) for NIR-PIT against recurrent HNSCC with Cetuximab-IR700, the median OS was 9.30 months (95% CI 5.16–16.92 months). Unconfirmed ORR was achieved in 13 (43.3%, 95% CI 25.46%–62.57%) patients, with 4 (13%) patients achieving CR and 9 (30.0%) patients demonstrating PR. Disease control was observed in 24 (80%, 95% CI 61.43%–92.29%) patients [37]. Theoretically, EGFR-targeted NIR-PIT is applied to any type of cancer in which EGFR is overexpressed, and clinical trials are expected to progress in thoracic tumors.

### 4.2. HER2

HER2 is a membrane tyrosine kinase receptor and, together with EGFR, is a member of the ErbB family [25]. When overexpression of HER2 occurs, it forms homodimers or heterodimers with other ErbB family receptors; thus, activating oncogenic downstream signals that promote cell proliferation, survival, and angiogenesis [38]. Several antibody drugs, including trastuzumab, pertuzumab, and trastuzumab emtansine (T-DM1), have been approved by the FDA for the treatment of breast cancer, with HER2 positivity rates of 15–20% [39]. In addition, trastuzumab and fam-trastuzumab deruxtecan-nxki have already been approved by the FDA for the treatment of gastric cancer, where approximately 20% of cases are HER2 positive. These have been applied as mAbs in NIR-PIT; NIR-PIT with trastuzumab-IR700 showed valid results in a pleural dissemination model with HER2-expressing NSCLC cells and a lung metastasis model with HER2-expressing 3T3 cells [40,41,42]. In addition, it has been reported that chemo drugs, especially cisplatin-resistant (SBC-3/CDDP) cell lines upregulate HER2 expression. NIR-PIT against HER2 is effective also against tumors in patients who have already become resistant to chemotherapy [43].

### 4.3. CD44

CD44, a non-kinase transmembrane glycoprotein, regulates intercellular adhesion and epithelial–mesenchymal transition in normal cells. At the same time, it is a marker for the identification of cancer stem cells (CSCs), as CD44 increases tumor development and progression. Various types of cancer, including lung cancer, express CD44, which has been shown to be a poor prognostic factor [44,45,46]. Therefore, CD44 is an important approach for antibody therapeutics to eliminate CSCs [47]. In NIR-PIT targeting CD44 positive oral squamous cell carcinoma and breast cancer in a mouse model, tumor progression was significantly inhibited and survival was prolonged [48,49]. Additionally, in immunocompetent mouse models, the effect was further enhanced combined with immune activation using type 1 cytokine or immune checkpoint inhibitors [50,51,52].

### 4.4. CEA

Carcinoembryonic antigen (CEA), a glycoprotein involved in cell adhesion, is highly expressed in many epithelial cells of tumors, including lung adenocarcinomas. It has already been used as a tumor marker for various cancers, and its expression level has been associated with the prognosis of patients with colorectal cancer [53,54,55]. NIR-PIT targeting CEA significantly inhibited tumor progression without side effects in xenograft models of gastric cancer and orthotopic pancreatic tumor models [56,57]. In pancreatic cancer, the use of NIR-PIT as an adjunct to surgery to treat residual pancreatic cancer in a patient-derived orthotopic xenograft model also was found to improve overall survival [58,59].

CEA is also expressed in normal cells, mostly derived from surface cells of the colon, which are released into the intestinal tract and excreted in the stool [56]. Therefore, even if stray NIR light irradiated to the chest reaches normal colonic tissue, it is suggested that the side effects are very minimal. Thus, CEA is a promising target for the treatment of thoracic cancer.

## 5. Exploring Other Targets to Detect More Specific and Effective Markers for Lung NIR-PIT: PDPN, MSLN, GPR87, and DLL3

### 5.1. PDPN

Podoplanin (PDPN) is a type I transmembrane glycoprotein expressed in lymphatic endothelial cells, type I alveolar epithelial cells, and glomerular podocytes. Antibodies against PDPN (D2-40) have long been used as specific pathological diagnostic markers to confirm the presence of MPM [60,61]. PDPN has been reported to be expressed in various tumors, including MPMs [62,63]. NIR-PIT, which targets PDPN with NZ-1 antibody, is a promising target against MPM and exhibits tumor-suppressive effects in both xenograft and orthotopic MPM models [64]. It has also been shown to be effective in an orthotopic mouse model of pleural disseminated lung cancer [65].

### 5.2. MSLN

Mesothelin (MSLN) is a cell surface glycoprotein and a tumor differentiation marker expressed in multiple tumors, including lung cancer and MPM [66]. The association between MSLN overexpression and poor survival in MPM has not been consistently reported [67,68]. However, in MPM, MSLN is known to be associated with epithelial–mesenchymal transition (EMT) and binding to mucin 16 (MUC16/CA125), which is associated with cancer progression and aggressiveness [69,70]. MSLN is highly expressed in tumors, while its expression in normal cells is limited, making it a promising biomarker and therapeutic target [71,72]. NIR-PIT targeting MSLN was effective in a mouse xenograft model [73]. Thus, NIR-PIT targeting MSLN holds good potential for the treatment of tumors expressing mesothelin.

### 5.3. GPR87

GPR87 is a G-protein receptor that is highly expressed specifically in both lung cancer and MPM and is rarely expressed in normal cells [74,75]. GPR87 is a poor prognostic factor [75,76,77]. NIR-PIT targeting GPR87 produced a therapeutic effect in a mouse model with transplanted cell lines of lung adenocarcinoma, SCLC, and MPM [78]. GPR87 may be an ideal target for treating thoracic tumors.

### 5.4. DLL3

Delta-like protein 3 (DLL3) is a ligand for the Notch receptor and a promising therapeutic target molecule for small cell lung cancer (SCLC) and other neuroendocrine tumors [79,80]. Its expression is rarely observed in normal tissues. Rovalpituzumab, a DLL3-targeted antibody drug, has already been used in human clinical trials [81]. NIR-PIT targeting DLL3 with rovalpituzumab in an SCLC xenograft model has shown significant anti-tumor effects, suggesting that DLL3 is a promising target [82].

### 5.5. CD26

CD26 is a type II transmembrane protein that recognizes chemokines with the penultimate proline or alanine and cleaves the NH2-terminal dipeptide [83,84]. Immunohistochemical analysis has shown that CD26 is highly expressed in MPM cells, indicating its pro-carcinogenic function and relevance as a prognostic marker [85]. Promising results have also been obtained with antibody therapy targeting CD26 in phase I clinical trials [86]. It is expected that NIR-PIT will be indicated for MPM.

### 5.6. CDH3

CDH3 is a calcium-dependent adhesion molecule and a member of the cadherin superfamily [87]. It is overexpressed in advanced lung adenocarcinoma and has been reported to be associated with poor prognosis and EGFR-TKI resistance [88,89].

### 5.7. TROP2

Trophoblast surface antigen 2 (Trop2) is a widely expressed transmembrane glycoprotein and a member of the epithelial cell adhesion molecule (EpCAM) family [90]. It is highly expressed in a variety of epithelial tumors, including NSCLC, and has been reported to be associated with prognosis and as a potential new therapeutic target [91,92].

### 5.8. XAGE1

XAGE1 is a member of the cancer/testis (CT) antigen family expressed in a variety of cancers including NSCLC and in testicular germ cells [93]. CT antigens are widely expressed in cancers but rarely in normal cells and are highly immunogenic, making them promising targets for immunotherapies such as cancer vaccines and NIR PIT [94,95,96].

## 6. Immunogenic Targets to Detect More Specific and Effective Markers for Lung NIR-PIT: CD25, PD-L1, CTLA4

Although the tumor microenvironment(TME) is rich in T cells and natural killer (NK) cells that can recognize cancer cells, the presence of immunosuppressive cells such as regulatory T cells (Tregs) in the vicinity acts as a mechanism to evade their cytotoxic function [97]. Immune checkpoint (ICP) molecules are also known to inhibit anti-tumor immune reactions. The regulation of immunosuppressive cells or ICP in the tumor microenvironment is an important step for enhancing anticancer immune responses. Spatiotemporal decrease in Tregs or modulating ICP could augment anti-tumor immune reactions and inhibit tumor growth. Therefore, immunosuppressed cells within the TME are a promising target for NIR-PIT cancer therapy.

### 6.1. CD25

CD25 is the undisputed IL-2 receptor mainly expressed on activated Tregs but not on naive T cells, and NIR-PIT targeting this receptor has been developed to manipulate the cellular environment of the tumor microenvironment [98].

Tregs are highly immunosuppressive, and a higher proportion of Tregs in tumor-infiltrating lymphocytes (TILs) is associated with a poorer prognosis; in HNSCC, the degree of Treg infiltration in the TME correlated with prognosis [99,100]. Systemic administration of simple anti-CD25-IgG has been reported to deplete peripheral Tregs [98] and may also induce autoimmune adverse events such as cytokine storms. However, NIR-PIT can selectively reduce only Tregs in the TME without removing local effector T cells or Tregs present in other organs by precise irradiation.

NIR-PIT targeting CD25 caused necrotic cell death in CD25-expressing helper T cell lines in vitro and had no effect on cancer cells. However, in an in vivo syngeneic mouse model, it induced regression of the treated tumors with rapid activation of CD8+ T cells and NK cells infiltrating the tumors and activation of antigen-presenting cells. CD25-targeted NIR-PIT selectively depleted Tregs from TME, resulting in activation of effector cells and upregulation of anti-tumor immunity. Furthermore, CD8+ T cell and NK cell activation and antitumor effects were also observed in non-irradiated, distant, non-treated tumors (abscopal effect) [97]. Local CD25-targeted NIR-PIT may be a safer alternative to systemic Treg depletion because it depletes Tregs only at the tumor site where the NIR light is irradiated. The effect of Treg depletion by NIR-PIT lasted for 3–4 days, followed by gradual repopulation of Tregs, reaching the pre-treatment number of Tregs approximately 6 days after treatment [97]. If the tumor recurs, NIR-PIT can be repeated, and repeated depletion of Treg cells by CD25-targeted NIR-PIT can prolong tumor control and survival. Combination NIR-PIT dramatically improved the CR rate in a syngeneic mouse model [101].

### 6.2. PD-L1

Programmed death ligand 1 (PD-L1) is an immune checkpoint that binds to PD-1 (CD279) on T cells to reduce immune response cytokines and induce immune tolerance in tumor cells [102,103]. It is overexpressed in many cancer cells, including lung cancer, and is associated with poor prognosis by inhibiting T-cell immune responses [104].

NIR-PIT against PD-L1 is effective even when tumor PD-L1 expression is low and is suitable for tumors in a variety of patients, regardless of the type or organ. It is also therapeutically effective against tumors in remote areas that are not directly irradiated, even though NIR-PIT causes minimal side effects in areas that are not irradiated. In addition, in the inflammation of tumor cells caused by photocytotoxicity, cytokines such as INF-γ can promote the expression of PD-L1 and activate PD-L1 in treated tumor cells. Therefore, NIR-PIT targeting PD-L1 provides additional benefits from repeated PD-L1-targeted therapy.

NIR-PIT with antibodies against PD-L1 significantly inhibited tumor growth and prolonged survival in xenograft models by activating CD8+ T cells and NK cells in the tumor microenvironment by promoting PD-L1 positive cell rupture and inhibiting tumor immunosuppressive pathways [105,106].

### 6.3. CTLA4

Cytotoxic T-lymphocyte antigen-4 (CTLA4) is an immune checkpoint protein expressed on regulatory T (Treg) cells and activated T cells that downregulate T cell activation and suppress anti-tumor immune responses in response to T cell receptor engagement [107,108,109]. Local depletion of CTLA4 expressing cells by NIR-PIT promotes the activation and infiltration of CD8+ T cells in the tumor microenvironment and prolonged survival in vivo [110] with observation of CD8+ T cell activation and infiltration. Additionally, the same as CD25, combination NIR-PIT dramatically improves the CR rate in syngeneic mouse models [111].

## 7. NIR Light Irradiating Devices

NIR light significantly attenuates through deep and hard tissues by depending on the light penetration limit. Additionally, NIR-PIT against immunogenic targets in the normal lung tissue can cause an acute respiratory distress syndrome (ARDS)-like reaction, although it has already been proven that the one against the target of tumor antigen, such as HER2, does not adversely affect normal tissue. The precise irradiation and evaluation of NIR light irradiation are required. For this reason, an optical fiber diffuser in the form of a flexible cylindrical optical fiber was proposed [112,113,114,115,116,117]. The optical fibers are utilized through the bronchoscope to deliver NIR-light to key areas. This bronchoscope technique can be navigated with a 3D navigation system from CT images beforehand to guide the fiber precisely to the tumor in the lung. With these techniques, damage to the normal lung fields could be minimized. In addition, an implantable wireless NIR light-emitting diode has been developed that can be implanted once and used for irradiation several times [118]. Treatment with a single dose of high-energy NIR light can injure even normal tissues; however, multiple doses at the recommended energy of 500 mW/cm2 (at 690 nm) or less can produce anti-tumor effects. By combining this with an endoscope or catheter, it is possible to deliver light locally within the body without burns or other injuries [119,120,121,122].

## 8. Imaging Modality

Visualization of the NIR-PIT irradiation site and monitoring of the treatment effect at that site in real time or immediately after treatment are important to determine if accurate treatment is being achieved and if additional treatment is needed.

Fluorescence imaging of the IR700 can confirm the distribution of antibody–photoabsorber conjugates, and the NIR laser light irradiation causes the IR700 to photobleach; thus, reducing fluorescence during treatment. In the preclinical setting, the fluorescence wavelength of the IR700 (720 nm) could be measured, but a new, dedicated measurement device would have to be installed to handle it in the clinical setting. In addition, preclinically, bioluminescence imaging using luciferase can be used to evaluate the therapeutic effect of NIR-PIT after treatment, but clinical implementation is not possible [123].

Taking advantage of the broad emission spectrum of IR700, a clinically approved camera designed to detect indocyanine green, typically at 830 nm, was proposed for use during NIR-PIT, the distribution of previous IR700, and the progression of photochemical reactions can be monitored in real time and would allow optimization of NIR light irradiation during NIR-PIT [124,125]. In addition, there have been attempts to evaluate treatment efficacy by assessing the SUPR effect with the administration of indocyanine green (ICG) particles. Previous studies have shown that dynamic fluorescence imaging with ICG shows increased signal intensity in tumors after NIR-PIT treatment; significantly higher ICG intensity was demonstrated from NIR-PIT-treated tumors as early as 20 min after ICG injection [126], indicating the possibility of using SUPR effect assessment by ICG imaging to evaluate the acute cytotoxic effects of NIR-PIT [124,125,127]. Magnetic resonance imaging (MRI) imaging using gadofosveset also showed gradual signal enhancement up to 30 min in NIR-PIT-treated tumors, suggesting that it may be a useful imaging biomarker for detecting treatment changes after NIR-PIT [128].

Tumors require excess glucose due to the Warburg effect, which dramatically increases the rate of glucose uptake [129,130]. ^18^F-fluorodeoxyglucose positron emission tomography (^18^F-FDG PET) has been used to exploit this, for example, for tumor detection [131,132,133]. ^18^FDG-PET is useful as a rapid response marker of therapeutic response, as glucose metabolism in treated tumors was greatly reduced early after NIR-PIT [134].

## 9. Perspective on ADCs (Antibody–Drug Conjugates)

The mAb carriers used in ADCs, which aim to reduce the impact of toxic chemical cancer drugs on normal tissues, always have inherent off-target toxicity. Even very small amounts of mAb–drug conjugates reaching epitopes in normal cells are considered to be highly toxic; therefore, lowering the therapeutic index. The application of ADCs in NIR-PIT studies has been reported to enhance the antitumor effect of the T-DM1-IR700 conjugate and the transport of NIR light-sensitive drug-releasing antibodies by the SUPR effect [135,136,137]. The mAb used in this conjugate was the first to reach clinical implementation, and the IR700 added to the ADC caused the SUPR effect, which is an inherent nanoparticle accumulation enhancement phenomenon occurring after NIR-PIT, to accumulate the antibody carrier in the tumor area, followed by NIR light-triggered release to expose the drug only to a specific area. The fluorescence signal evaluation confirmed the accumulation of the antibody carrier and the released anticancer drug in the tumor area, indicating an additional antitumor effect of the anticancer drug. The active and potent agent is produced only at the irradiated site, whereas the non-irradiated, non-targeted mAb–dye conjugate is essentially non-toxic. Therefore, they can be used to treat cancer as a secondary effect of each other with reduced toxicity, without compromising specificity or their respective advantages [105].


**Figure 1 biomedicines-10-01662-f001:**
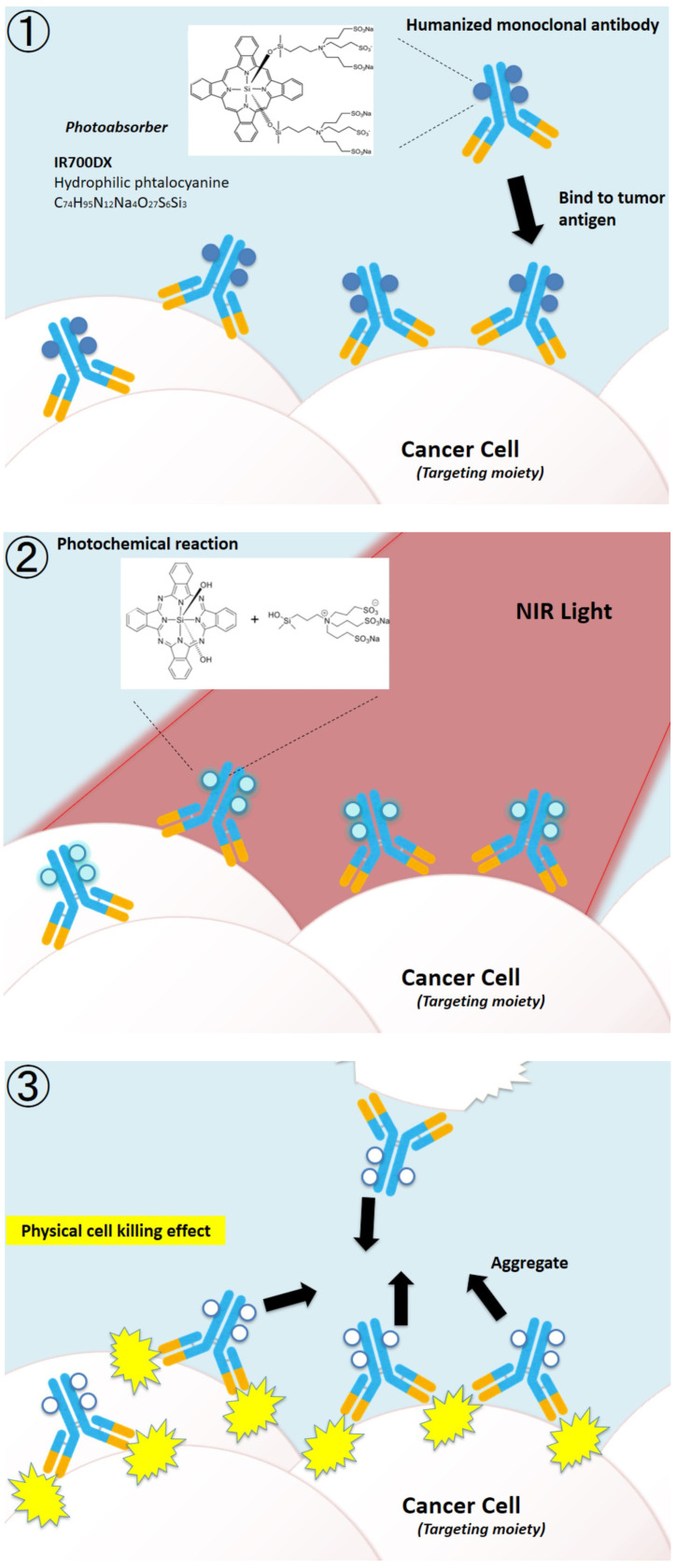
Schematic representation of near-infrared photoimmunotherapy. The conjugates, consisting of humanized monoclonal antibody and photo absorber IR700DX, can target specific cancer cell antigens. Once NIR-light irradiates, the targeting cancer cells are ruptured.

## 10. Conclusions

NIR-PIT overcomes the side effects that used to be a problem with anticancer drugs and photodynamic therapy, and has achieved a new, ultra-specific, multidisciplinary therapy that can approach cancer treatment directly and immunologically at the cancer site. Not only that, it has the potential to be widely applicable to any type of cancer by replacing antibodies.

NIR-PIT for thoracic cancer was not realistic because of the difficulty of external irradiation due to the surrounding ribs and the concern about ARDS symptoms caused by the ICD associated with the treatment. Recently, the development of uniform irradiation methods and modalities that enable rapid monitoring of treatment effects has made it possible to provide appropriate and effective treatment to localized thoracic tumors. NIR-PIT is a very promising treatment candidate for thoracic cancer.

## Figures and Tables

**Table 1 biomedicines-10-01662-t001:** Candidate targets for thoracic cancer.

	Target Antigens
Versatile targets	EGFR CD44 CEA HER2 GPR87
Exploring targets	PDPN MSLN GPR87 DLL3 CD26 CDH TROP2 XAGE1
Immune targets	CD25 PD-L1 CTLA4

## Data Availability

Not applicable.

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
