# Peer review of "Near-Infrared Photoimmunotherapy for Thoracic Cancers: A Translational Perspective"

_biomedicines, 2022, doi:10.3390/biomedicines10071662_

Round 1

Reviewer 1 Report

The review presented by K. Sato and coworkers reports an emerging specific cancer treatment: the near-infrared photoimmunotherapy (NIR-PIT) and its potential applications for thoracic cancers.

I suggest some minor corrections and some additional precisions:

Page 1, lines 37-39: This sentence highlights the NIR light that has maximal tissue penetration with minimal endogenous light absorption. The authors could cite un review concerning this topic. They might suppress ref 3 and 4, that report PDT and PTT studies with chlorin as a photosensitizer that is excited at 660 nm.

Page 1, line 43: ECLC: please develop the acronym.

Page 2, lines 47-49:  reference 6 uses HPD, a natural hematoporphyrin derivative, that is excited with red light at 630 nm. So this sentence should be reformulated, or other references corresponding to the data given in this sentence (lipid-soluble molecules, visible light) should be given.

Page 3, line 103: SUPR please develop the acronym

Page 7, line 329: ARDS please develop the acronym

Page 8, first paragraph: the authors could also add the irradiation wavelength (or wavelength range) used with these techniques. Indeed, in addition of the strength of the applied light (500 mW/cm2 for example), the information of the wavelength is also helpful.

Page 8, line 348: APC please develop the acronym

Page 8, line 349: replace “irradiation NIR laser light” by “NIR laser light irradiation”

Page 9, line 384: replace “NIR-release” by “NIR light-triggered release”

Page 9, Figure 1: in the figure, remove the protecting group of the photoabsorber (as it is absent once fixed on the antibody. Present simple arrows (as different arrow styles have different meaning). Add an arrow going to Targeting moiety.

Page 10, line 401: replace “has realized” by another verb.

Page 10, line 405 and 410: remove therapy as it is already present in the acronym.

Page 18, reference 116: complete this reference.

Author Response

Please see the attached rebuttal letter.

Reviewer 2 Report

Specific comments:

First of all, light of 690 nm wavelength is not IR. It is part of the visible range!

See any relevant  textbook or https://science.nasa.gov/ems/09_visiblelight, or https://www.britannica.com/science/color/The-visible-spectrum or many others.

Introduction is very confusing and superficial. Here I have to mention that even in “traditional PDT” (line 90) several delivery systems are used (liposomes, monoclonal antibodies, various nanoparticles, etc.)

It is not clear what authors believe about the role of the light in NIR-PM. How can light induce aggregation (I would say “binding”) of antibody and antigen proteins? How can light contribute to target selection? How can NIR-PIM “decrease the production of reactive oxidative species” (line 86). In this case what is the role of photosensitizer phthalocyanine derivative IR700?

It is not clear how NIR-PM can delivery of nanoparticles (line 115). It is not clear how nanoparticles come to the picture at all.

 NIR-PM is randomly written as NIR-PM or NIR PM.

Many abbreviations are not explained, e.g., ECLC (possibly early central lung cancer) or  SUPR (possibly super enhanced permeability and retention).

Chapter 5 and 6 have similar title (see line 193 and 259)

Reference numbers appear randomly in the paper, e.g., [100] (line 106) is coming after [10] (line 95).

In chapters 5.5., 5.6., 5.7., 5.8., do not refer any experimental results on NIR-PM. These chapters must be removed from the manuscript.

Author Response

(The authors gave the same response as above.)

Reviewer 3 Report

In this manuscript, the authors summarize the recent status and the possible perspective of the near-infrared photoimmunotherapy stategy for thoracic tumors applications. The manuscript is well written and the conclusions are interesting. The major weakness of the manuscript lies in its being discursive and hard to understand for non-specialists. Adding a few figures and tables can significantly improve the work presented.

Author Response

@lease see the attched pdf file.
